# Real-World Practices of Pentosan Polysulfate Maculopathy Screening in Various Clinical Settings and Practice-Associated Factors

**DOI:** 10.3390/jcm13175090

**Published:** 2024-08-27

**Authors:** Jiyeong Kim, Seong Joon Ahn

**Affiliations:** 1Department of Pre-Medicine, College of Medicine, and Biostatistics Lab, Medical Research Collaborating Center (MRCC), Hanyang University, Seoul 04763, Republic of Korea; kimzi@hanyang.ac.kr; 2Department of Ophthalmology, Hanyang University Hospital, Hanyang University College of Medicine, Seoul 04763, Republic of Korea

**Keywords:** pentosan polysulfate, maculopathy, screening practice

## Abstract

**Objectives**: This study investigated the practice patterns of pentosan polysulfate (PPS) maculopathy screening in various clinical settings and demographic and clinical characteristics associated with these screening practices using a health claims database. **Methods**: In this nationwide population-based study, data from the Health Insurance Review and Assessment database in South Korea were analyzed to identify patients who underwent PPS. The participants were categorized based on whether they had undergone a baseline examination (the first ophthalmic examination since PPS prescription) within one year of PPS use, subsequent monitoring within one year of the baseline examination, or recent monitoring within a 1-year period before the study end date. Demographic and clinical factors were compared between the groups, and factors associated with screening practices were identified using logistic regression analyses. **Results**: Significant differences in screening practices were observed based on sex, age, residence, the medical specialty of the prescribing physician, indications for PPS use, and hospital type of prescription. Older patients who received PPS prescriptions from urologists were more likely to undergo baseline and monitoring examinations. Logistic regression analyses revealed that older age, female sex, and a longer duration of PPS use were significantly associated with baseline screening. Subsequent and recent monitoring was significantly associated with age, duration of PPS use, and treatment at primary hospitals. **Conclusions**: This study underscores the variability in screening practices for PPS users based on demographic and clinical factors, emphasizing the need for standardized guidelines. Enhanced awareness and timely referral for maculopathy screening, particularly among non-urological specialties, are essential.

## 1. Introduction

Pentosan polysulfate sodium (PPS) is primarily used to treat interstitial cystitis, a chronic bladder condition that causes pain and urinary symptoms such as frequent urination, urgency, and irritation [1,2,3]. Recently, PPS has been linked to a specific type of maculopathy, known as PPS maculopathy, prompting concern over its long-term ocular safety [4,5,6]. PPS maculopathy is characterized by retinal pigment epithelium (RPE) dysfunction and progressive degeneration, often presenting as pigmentary changes and macular atrophy even after drug discontinuation [4,5,6]. The proposed pathophysiology involves RPE toxicity, possibly associated with disruption of the interphotoreceptor matrix [7]. This condition is critical because it can lead to permanent vision loss [4,7]. The prevalence of PPS-associated maculopathy is increasing, and studies have indicated a potential risk in chronic PPS users [7,8]. Advanced imaging modalities, such as optical coherence tomography (OCT), near-infrared reflectance (NIR), and fundus autofluorescence (FAF), facilitate the early detection of this condition, enhancing the ability to prevent significant visual impairment through timely drug cessation [7].

Despite the availability of these technologies, screening guidelines for PPS maculopathy have not been well established, which may lead to missed detections and diagnoses. Some authors recommend an initial baseline eye examination for patients starting PPS therapy, followed by annual eye examinations using multiple imaging techniques, such as OCT, FAF, and NIR, for those with chronic PPS exposure (e.g., cumulative dosages exceeding 500 g) [7,9]. However, clinicians’ awareness of this disease entity and its screening is variable, and screening practices may widely vary in different settings. Few studies have assessed real-world screening practices; thus, screening practices in various specific clinical settings and the factors affecting these practices remain unknown [10,11]. However, identifying the factors influencing these practice patterns and those associated with insufficient screening practices is essential to target specific populations and guide clinicians in better implementing current recommendations for maculopathy screening [7].

This study aimed to explore screening practices among patients administered with PPS in diverse clinical settings, along with the demographic and clinical factors associated with retinopathy screening practices, to provide insights into real-world practice patterns for PPS maculopathy. Accordingly, we investigated the demographic and clinical characteristics associated with PPS maculopathy screening practices in a nationwide population-based cohort in Korea. By reflecting real-world practice patterns across all primary and referral centers at the national level, this study addresses the factors associated with practice patterns, which should be considered to enhance these practices.

## 2. Materials and Methods

### 2.1. Subjects

This nationwide, population-based study used data from the national Health Insurance Review and Assessment (HIRA) database recorded on 1 January 2015, and 31 December 2021. From the database, we identified and included PPS users between 1 January 2015 and 31 December 2021 for this study. All prescriptions and procedures in this mandatory universal health insurance system, which provides medical care to nearly the entire South Korean population (97%), are recorded in the HIRA database, as described previously [12,13]. Patients administered PPS were identified by searching for those with the drug component codes for PPS (code: 210701ACH).

To ensure a precise estimation of the duration of PPS use, we excluded patients who had previously received PPS, as these individuals might have been prescribed PPS prior to the inclusion period; thus, preventing an accurate estimation of the duration of PPS use. Additionally, we excluded patients examined with fundoscopy or any of the sensitive modalities recommended for screening PPS maculopathy such as OCT and FAF, and who were diagnosed with prior ophthalmic diseases. Furthermore, we excluded patients diagnosed with diabetes mellitus (DM) during the study period as an ophthalmologic screening, if performed, could be performed for diabetic retinopathy. Accordingly, this study included at-risk patients without prior ophthalmic disease or DM (requiring follow-up ophthalmic examinations or regular retinopathy screening) who started PPS therapy from 2018. Figure 1 provides a flowchart indicating the inclusion and exclusion criteria and the number of patients meeting these criteria. This study was approved by the Institutional Review Board of Hanyang University Hospital and was conducted in accordance with the principles of the Declaration of Helsinki.

### 2.2. Definitions

Two types of screening examinations: baseline examination and subsequent monitoring, were defined as reported previously [11]. For the baseline examination, previous reports recommended that all patients beginning PPS therapy undergo a baseline ophthalmologic examination within the first year of starting PPS. For subsequent monitoring examinations, previous recommendations suggested annual follow-ups [7], for which we defined a “1-year interval” as appropriate subsequent monitoring. We evaluated subsequent monitoring examinations regarding those involved within 1-year following the baseline examination and within a 1-year interval before the study end date (31 December 2021). Regarding screening modalities, previous reports recommended OCT, near-infrared (NIR) imaging, FAF, and fundus examination for structural tests, and automated visual field (VF), and multifocal electroretinogram (mfERG) for functional tests [7,9]. However, the HIRA database does not include a code for NIR; thus, it could not be evaluated in our analyses.

Thus, we defined baseline screening as the first ophthalmic examination performed for PPS users using fundoscopy/fundus photography, and/or OCT, FAF, automated VF, and mfERG. Subsequent monitoring was defined as examinations performed after the baseline examination using any of the above modalities, while recent monitoring was defined as those performed within the 1-year period before the study end date.

### 2.3. Evaluations

We evaluated the timing and modalities used for baseline screening and annual monitoring. We separated the patients into subgroups based on performance in the baseline examination and subsequent monitoring. We also divided the patients into subgroups based on the number of modalities used for monitoring examinations (single or multiple modalities). We compared demographic and clinical characteristics between patients with and without baseline examinations, and between those with and without monitoring. These characteristics included age, sex, residence, medical specialties prescribing PPS, hospitals (level of care) providing prescriptions, hospitals performing maculopathy screening, medical indications for PPS use, daily doses, and duration of PPS use.

### 2.4. Analyses

Categorical variables are reported as frequencies and percentages, whereas continuous variables are reported as means ± standard deviation (SD). We also report the median for the time period due to its skewed (non-normal) distribution. Based on the normality of the data, as determined by the Shapiro–Wilk test, either Student’s *t*-test or a Mann–Whitney U test was used to compare continuous variables between independent groups. We applied the chi-squared test to compare categorical variables between the groups. Multivariate logistic regression was performed to identify demographic and clinical factors associated with baseline performance and monitoring examinations. Odds ratios (ORs) were used to quantify the strength of the association between variables with 95% confidence intervals (CIs) provided as a measure of the effect size. We determined the statistical significance using a two-sided test with a significance level of 0.05. SAS Enterprise Guide 7.1 (SAS Institute, Cary, NC, USA) was used for all analyses.

## 3. Results

### 3.1. Demographic and Clinical Characteristics

Table 1 provides an overview of the demographic and clinical characteristics of the 58,752 patients who underwent PPS included in this study. The mean age was 56.0 years. Most patients taking PPS were aged ≥60 years (44.9%), followed by those aged 50–59 (21.8%), 40–49 (15.2%), 30–39 (10.2%), and <30 years (7.9%). The sex distribution was skewed towards women, who comprised 61.8% (*n* = 36,287) of the cohort, compared with men at 38.2% (*n* = 22,465).

Regarding residence, the population was almost evenly split between metropolitan/large cities (47.5%) and small cities/rural areas (52.5%). Urologists were the primary prescribers of PPS, accounting for 86.4% of prescriptions, followed by obstetricians/gynecologists (OB/GYNs, 8.7%), internal medicine specialists (2.6%), and other specialties (2.3%). A substantial proportion of the prescriptions were made at primary hospitals (54.4%) compared with referral centers (45.6%). The medical indications for PPS use were predominantly interstitial cystitis (92.9%), other forms of cystitis (2.8%), and other conditions (4.3%). The mean duration of PPS use was 3.5 months, and mean daily dose was 246.4 mg. Among dosing categories, 12.4% of users took <200 mg daily, 37.4% took between 200–299 mg, and 50.3% took ≥300 mg.

### 3.2. Practices and Characteristics of Baseline and Monitoring Examinations across Different Settings

Table 2 outlines the percentages of patients taking PPS who received a baseline examination within 1-year post-PPS initiation, categorized according to the demographic or clinical settings and characteristics. The results showed that 14.0% of patients in metropolitan or large cities and 13.9% in small cities or rural areas had undergone baseline examinations, with no significant difference between these groups (*p* = 0.976). Patients prescribed PPS by urologists had the highest rate of baseline examinations (14.4%), followed by those prescribed by internal medicine specialists (13.4%), other specialties (12.9%), and OB/GYNs (10.3%, *p* < 0.001). Additionally, baseline examinations were significantly more common in primary hospitals (15.1%) compared with referral centers (12.6%, *p* < 0.001). Regarding the medical indications for PPS use, the rates of baseline examinations were similar across patients with interstitial cystitis (14.0%), other cystitis (13.9%), and other indications (13.6%, *p* = 0.838).

Table 3 presents the percentages of patients taking PPS who received subsequent monitoring within 1 year after a baseline examination and recent monitoring during 2021 according to settings and characteristics. The data indicated a significant difference between the proportions of patients in large cities and small cities or rural areas that received subsequent monitoring within 1 year (8.9 vs. 8.5%, *p* = 0.003). Additionally, a significantly higher proportion of patients in small cities or rural areas had received recent monitoring compared with large cities (13.5 vs. 12.5%, *p* < 0.001).

Regarding the medical specialties prescribing PPS, urology patients had the highest rates of subsequent (9.0%) and recent (13.2%) monitoring. Patients prescribed PPS by OB/GYNs had the lowest rates of subsequent (5.8%) and recent (10.7%) monitoring, with significant differences observed in recent monitoring (*p* = 0.001). Patients treated in primary centers were significantly more likely to receive both subsequent (9.7%) and recent (13.5%) monitoring compared with those treated in referral centers (7.6 and 12.3%, respectively, both *p* < 0.001). The percentages of patients receiving subsequent monitoring within 1 year following baseline examination did not differ significantly between those receiving baseline examinations at primary (36.9%) and referral (36.7%) centers. However, the proportion of patients receiving recent monitoring was significantly higher among those receiving baseline examinations at referral centers compared with primary hospitals (55.3 vs. 51.2%, *p* = 0.001). Regarding medical indications for PPS use, patients with interstitial cystitis had subsequent and recent monitoring rates of 8.7 and 13.1%, respectively, with no significant differences among the various indications.

### 3.3. Comparison of Demographic and Clinical Factors According to Baseline and Monitoring Examinations

Figure 2 shows the results of comparing demographic and clinical factors between patients who underwent a baseline examination within 1 year of PPS initiation and those who did not. Patients with a baseline exam within 1 year were significantly older than those who did not (mean age, 60.9 years vs. 55.2 years, *p* < 0.001). The mean duration of PPS use was also longer in the baseline exam group compared with the no-baseline exam group (4.3 months vs. 3.4 months, *p* < 0.001). Regarding the mean daily dose, the baseline exam group had a slightly, but significantly, lower mean dose compared with the no baseline exam group (*p* < 0.001). Regarding sex distribution, a higher percentage of women underwent a baseline exam within 1 year compared with men (14.4 vs. 13.3%, *p* < 0.001).

Some demographic and clinical factors differed significantly between patients who did and did not undergo subsequent monitoring within 1 year after a baseline examination (Figure 3B). Patients who underwent subsequent monitoring had a mean age of 62.4 years, whereas those who did not had a mean age of 58.1 years (*p* < 0.001). The mean duration of PPS use was longer for patients with subsequent monitoring compared with those without (5.0 vs. 4.3 months, *p* < 0.001). The mean daily dose was slightly but significantly lower for patients who underwent subsequent monitoring compared with those who did not (242.3 vs. 246.4 mg, *p* < 0.001). Regarding sex distribution, a significantly higher proportion of men received subsequent monitoring within 1 year following baseline examination compared to that in women (38.9 vs. 35.4%, *p* < 0.001).

Patients who had undergone recent monitoring were an average of 59.3 years of age, compared with 60.2 years among those who had not (*p* < 0.001, Figure 3A). The mean duration of PPS use was shorter for patients with recent monitoring (4.1 months) compared to those without (5.1 months; *p* < 0.001). The mean daily dose was slightly higher for patients with recent monitoring (245.9 mg) compared with those without (243.7 mg; *p* = 0.051). Recent monitoring was performed in a significantly higher proportion of female patients taking PPS compared with men taking PPS (52.9 vs. 55.9%, *p* = 0.001).

### 3.4. Comparisons of Characteristics According to the Modalities Used for Monitoring Examinations

Appendix A details the frequency of various tests performed during the monitoring examinations of patients taking PPS. Funduscopy or fundus photography was the most commonly used modality, performed in 98.0% of PPS users, whereas OCT and automated VF testing were used in 45.5% and 15.6% of cases, respectively. Fundus autofluorescence was utilized in 5.2% of the users, whereas mfERG was the least frequently used test, performed in 0.6% of cases. Other tests were performed in 9.1%.

Table 4 shows the results of the comparison of the sensitive imaging modalities used for monitoring PPS users and their association with demographic and clinical characteristics. While we observed no significant differences in mean age, sex, and residence between the groups, patients who were monitored or receiving PPS prescriptions at primary hospitals more frequently underwent multiple modalities for monitoring, OCT and FAF, than those at referral centers (*p* < 0.001). However, the mean daily dose or duration was not significantly different between those receiving OCT without FAF monitoring and those receiving OCT and FAF (both *p* > 0.05).

### 3.5. Factors Associated with Undergoing Baseline and Monitoring Examinations

Table 5 provides insights from the results of multivariate logistic regression analyses to identify factors associated with baseline examination within 1 year, as well as subsequent and recent monitoring for patients taking PPS. Age was positively associated with undergoing a baseline examination (OR 1.023, 95% CI [1.021–1.024]; *p* < 0.001), and women were more likely to receive a baseline examination compared with men (OR 1.279, 95% CI [1.215–1.346]; *p* < 0.001). Patients from small cities or rural areas were less likely to have a baseline examination compared with those from metropolitan areas (OR 0.928, 95% CI [0.885–0.973]; *p* = 0.002). Urology was associated with a higher likelihood of baseline examination compared with OB-GYN (OR 0.806, 95% CI [0.729–0.890]) or other specialties (OR 0.807, 95% CI [0.885–0.973]; *p* < 0.001). A longer duration of PPS use was positively associated with having a baseline examination (OR 1.012, 95% CI [1.009–1.016]; *p* < 0.001); however, the mean daily dose did not significantly affect this outcome (OR 1.000, 95% CI [1.000–1.000]; *p* = 0.800).

For subsequent monitoring within 1 year, age was also positively associated with monitoring likelihood (OR 1.023, 95% CI [1.020–1.026]; *p* < 0.001), whereas sex was not (OR 0.958 for women, 95% CI [0.888–1.034]; *p* = 0.272). While the mean daily dose was not significantly associated with baseline and monitoring examinations (all *p* > 0.05), the duration of PPS use was significantly associated with baseline examination (OR 1.012, 95% CI [1.009–1.016]) and recent monitoring (OR 0.985, 95% CI [0.980–0.989]; both *p* < 0.001).

## 4. Discussion

Our study results revealed diverse patterns in screening examination uptake across various settings and medical specialties. Moreover, the findings highlight the significant impact of patient demographics, geographical location, medical specialties, and the type of hospital for PPS prescription and screening. Based on the associated factors, we intend to suggest tailored monitoring strategies to enhance patient care and outcomes for those with characteristics associated with no or poorly performed screening.

Our analysis of 58,752 PPS users revealed important demographic and clinical characteristics that inform screening strategies. Most PPS users were older adults, particularly women, thus providing a target population for focused screening. Urologists were the primary prescribers of PPS, followed by OB/GYNs, internists, and other specialties, underscoring the pivotal role of urologists in patient referral for monitoring. Additionally, a significant proportion of prescriptions were made at primary hospitals, primarily for interstitial cystitis. Considering the demographics of older, middle-aged women and involvement of primary hospitals, targeted screening and monitoring, including referrals for maculopathy screening, are crucial for improving patient care and outcomes in PPS users.

Patients who underwent a baseline examination for PPS maculopathy within 1 year of PPS initiation were older than those who did not (average age 60.9 vs. 55.2 years). These patients also had a longer duration of PPS use and slightly lower mean daily dose. Women were more likely than men to receive a baseline examination. This trend is beneficial, as older patients are more prone to macular diseases, such as age-related macular degeneration [14], which are crucial to eliminate during differential diagnosis of PPS maculopathy [5,7,15,16]. The longer durations of PPS use further emphasized the need for timely baseline screening and robust follow-up monitoring to effectively manage the increased risk of PPS maculopathy.

The comparison of demographic and clinical factors between patients who received subsequent monitoring within 1 year after a baseline examination and those who did not revealed several trends (Table 3 and Figure 3B). Patients who underwent subsequent monitoring were generally older compared with those who did not receive follow-ups (average age 62.4 vs. 58.1 years). Additionally, these patients had a longer mean duration of PPS use and a slightly lower mean daily dose. These findings indicated that patients with older age and a longer history of PPS use were more likely to undergo subsequent monitoring, probably due to increased risks or the need for more proactive detection [7].

Conversely, patients who had undergone recent monitoring were somewhat younger and had a shorter duration of PPS use compared with those who had not received a recent follow-up. This trend implies that recent monitoring is more prevalent among patients who are younger and were relatively recently on PPS. The association between the duration of PPS use and recent monitoring might also suggest improved screening practices over time, consistent with previous findings on yearly uptake of ophthalmic screening in PPS users [11], as both findings indicate that a greater percentage of more recent PPS users were monitored. Nevertheless, these findings highlight the importance of tailored and ongoing monitoring strategies, particularly for long-term PPS users, to ensure timely, regular screening and address potential gaps in follow-up care for maculopathy, due to the higher risk of maculopathy.

The analysis of medical specialties revealed a disparity in the performance of baseline and monitoring examinations among PPS users. Urologists were significantly more likely to conduct baseline examinations compared with OB/GYNs, with urology showing the highest association between baseline examinations and subsequent monitoring. This trend may be attributed to greater awareness and adherence to screening guidelines among urologists, possibly due to more extensive training and organizational recommendations specific to their field [17,18,19]. In contrast, OB/GYNs had the lowest rates of baseline examination and subsequent monitoring, suggesting a need for increased efforts to enhance OB/GYN awareness and implementation of screening practices.

The disparity in screening practices among different specialties highlights a critical gap that could significantly impact both current and future clinical practices in maculopathy screening. Addressing this gap through targeted interventions could help standardize care, ensuring that all PPS users receive appropriate and timely screening, regardless of the prescribing medical specialty. For example, targeted educational initiatives and enhanced communication between prescribing and screening physicians are crucial to ensure that OB/GYNs are equally informed and proactive in referring patients for PPS-related maculopathy, ultimately improving patient care and outcomes. Furthermore, this research serves as a valuable resource for clinicians by underscoring the importance of cross-specialty education and collaboration to ensure ocular safety in PPS users. Incorporating these findings into clinical guidelines and healthcare policies across multiple specialties, similar to the approach taken with hydroxychloroquine retinopathy [20], could lead to more consistent and effective maculopathy screening practices.

## 5. Limitations

This study has several limitations, however. First, the retrospective study design and reliance on operational definitions from a comprehensive medical claims database may have introduced selection bias, as these definitions may not fully capture all aspects of PPS exposure and monitoring. Additionally, while we aimed to include diverse clinical settings, the limitations of our database restricted comprehensive analyses of PPS maculopathy patterns in various clinical scenarios. Moreover, our findings are specific to the South Korean population, and their generalizability to other regions with different healthcare systems and demographics remains uncertain. This study also faced challenges in defining at-risk patients for maculopathy screening, as we included the entire population with any PPS prescription. However, if retinal toxicity does not develop within a very short timeframe, patients with such brief exposure should be excluded from our analyses. As the threshold of exposure required to induce PPS maculopathy remains undetermined [10], this should be explored in future studies. Additionally, while we focused on ophthalmic examinations coded for PPS screening, we could not completely exclude the possibility that some exams were conducted for other reasons. Future research should address these limitations and validate our findings across diverse populations and healthcare settings to improve understanding and screening practices for PPS-related maculopathy.

## 6. Conclusions

In summary, the results of this study revealed significant variations in PPS maculopathy screening practices across different settings and medical specialties. These findings emphasize the importance of targeting older adults, particularly women, for baseline and ongoing monitoring. Urologists, who are more frequently involved in PPS prescription, demonstrated a higher adherence to screening practices compared with other medical specialties, indicating a need for enhanced education and guideline dissemination among those other specialties, particularly in OB/GYN. The observed patterns in screening practices highlight the need for tailored and proactive monitoring strategies. Although the effect sizes were relatively small, their implications for clinical practice should not be overlooked, particularly in guiding targeted interventions. Future research should validate findings across diverse populations and refine screening protocols to ensure early detection and management of PPS-related maculopathy. By improving awareness and implementing targeted screening practices, healthcare providers can better manage the risks of maculopathy associated with PPS therapy and enhance patient outcomes.

## Figures and Tables

**Figure 1 jcm-13-05090-f001:**
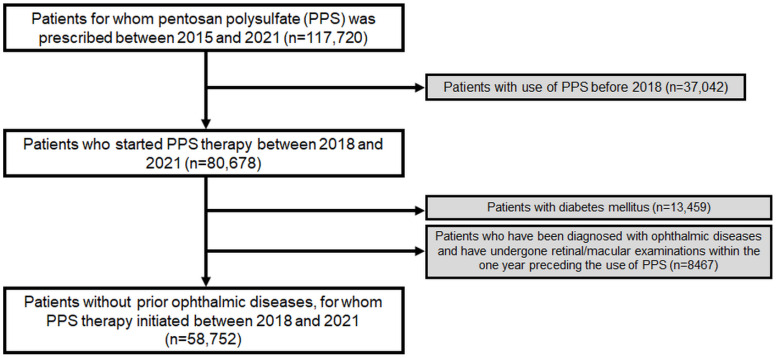
Flowchart of the study population inclusion and exclusion.

**Figure 2 jcm-13-05090-f002:**
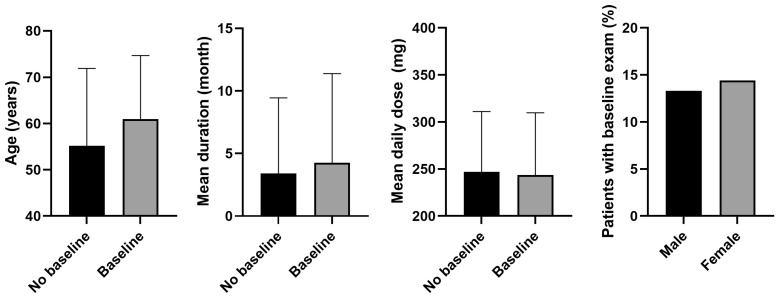
Comparison of demographic and clinical factors between patients who did and did not undergo baseline examination within 1 year of pentosan polysulfate (PPS) initiation. Significant differences were observed in all comparisons (*p* < 0.05). Error bars denote standard deviation.

**Figure 3 jcm-13-05090-f003:**
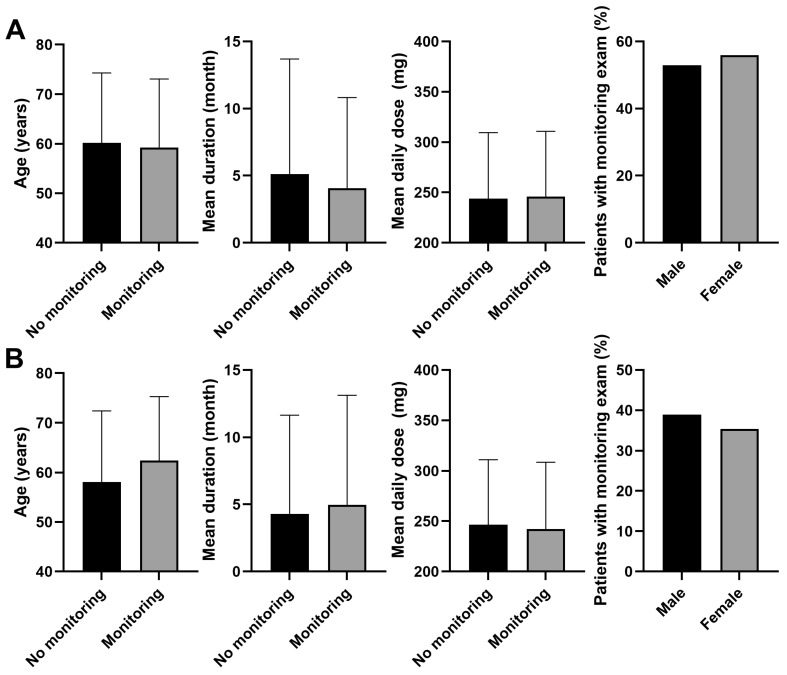
Comparison of demographic and clinical factors (**A**) between patients who underwent recent monitoring and those who did not, and (**B**) those who underwent subsequent monitoring within 1-year post-baseline examination and those who did not. Significant differences were observed in all comparisons (*p* < 0.05) except for the mean daily dose between those with and without recent monitoring (*p* = 0.051). Error bars denote standard deviation.

**Table 1 jcm-13-05090-t001:** Demographic and clinical information of patients using pentosan polysulfate (*n* = 58,752).

Characteristics	Mean or Value (SD or %)
Age, mean (years)	56.0 (16.5)
<30	4635 (7.9%)
30–39	6014 (10.2%)
40–49	8920 (15.2%)
50–59	12,815 (21.8%)
≥60	26,368 (44.9%)
Sex	
Male–female	22,465:36,287 (38.2%:61.8%)
Residence	
Metropolitan area/large cities–small cities/rural	27,890:30,862 (47.5%:52.5%)
Medical specialties prescribing PPS	
Urology	50,732 (86.4%)
Obstetrics/gynecology	5128 (8.7%)
Internal medicine	1548 (2.6%)
Others	1344 (2.3%)
Hospitals of prescription	
Primary–referral centers	31,965:26,787 (54.4%:45.6%)
Medical indications for PPS use	
Interstitial cystitis–Other cystitis–Others	54,597:1625:2530 (92.9%:2.8%:4.3%)
Mean/median duration, months	3.5/1.0 (6.2)
Daily dose (mg), mean	246.4 (64.5)
<200:200–299:≥300	7283:21,949:29,520 (12.4%:37.4%:50.3%)

**Table 2 jcm-13-05090-t002:** Percentages of patients using pentosan polysulfate (PPS) receiving a baseline examination within 1 year post-PPS initiation across settings and characteristics.

Settings/Characteristics	Percentage of Patients with Baseline Exam within 1 Year (%)	*p*-Value
Residence		
Metropolitan area/large cities	3891/27,890 (14.0%)	0.976
Small cities/rural	4303/30,862 (13.9%)	
Medical specialties prescribing PPS		
Urology	7284/50,732 (14.4%)	<0.001
Obstetrics/gynecology	529/5128 (10.3%)	
Internal medicine	208/1548 (13.4%)	
Others	173/1344 (12.9%)	
Hospitals of prescription		
Primary	4830/31,965 (15.1%)	<0.001
Referral centers	3364/26,787 (12.6%)	
Medical indications for PPS use		
Interstitial cystitis	7626/54,597 (14.0%)	0.838
Other cystitis	225/1625 (13.9%)	
Others	343/2530 (13.6%)	

**Table 3 jcm-13-05090-t003:** Percentages of patients using pentosan polysulfate (PPS) receiving subsequent monitoring within 1 year after baseline examination and recent monitoring during 2021, across settings and characteristics.

Settings/Characteristics	Percentage of Patients with Subsequent Monitoring within One Year (%)	*p*-Value	Percentage of Patients with Recent Monitoring (%)	*p*-Value
Residence				
Small cities/rural (Residence)	2368/27,890 (8.5%)	0.003	3769/27,890 (13.5%)	<0.001
Large cities	2745/30,862 (8.9%)		3860/30,862 (12.5%)	
Medical specialties prescribing PPS				
Urology	4572/50,732 (9.0%)	0.072	6708/50,732 (13.2%)	0.001
Internal medicine	130/1548 (8.4%)		199/1548 (12.9%)	
Obstetrics and gynecology	295/5128 (5.8%)		548/5128 (10.7%)	
Others	116/1344 (8.6%)		174/1344 (12.9%)	
Hospitals of prescription				
Primary	3090/31,965 (9.7%)	<0.001	4330/31,965 (13.5%)	<0.001
Referral centers	2023/26,787 (7.6%)		3299/26,787 (12.3%)	
Hospitals of baseline examinations				
Primary	694/1880 (36.9%)	0.847	962/1880 (51.2%)	0.001
Referral centers	4419/12,046 (36.7%)		6667/12,046 (55.3%)	
Medical indications for PPS use				
Interstitial cystitis	4777/54,597 (8.7%)	0.890	7126/54,597 (13.1%)	0.498
Other cystitis	134/1625 (8.2%)		195/1625 (12.0%)	
Others	202/2530 (8.0%)		308/2530 (12.2%)	

**Table 4 jcm-13-05090-t004:** Sensitive imaging modalities used for monitoring and their associations with demographic and clinical characteristics.

Characteristics	OCT without FAF (*n* = 4841)	OCT with FAF (*n* = 443)	*p*-Value
Mean age (years)	62.5 ± 13.1	61.6 ± 12.2	0.139
Sex			
Male Female	1902 (91.5%) 2939 (91.7%)	176 (8.5%) 267 (8.3%)	0.856
Residence			
Metropolitan area/large cities Small cities/rural	2276 (91.7%) 2565 (91.5%)	206 (8.3%) 237 (8.5%)	0.836
Medical specialties prescribing PPS			
Urology Internal medicine Obstetrics and gynecology Others	4280 (91.6%) 136 (90.7%) 307 (93.0%) 118 (91.5%)	395 (8.5%) 14 (9.3%) 23 (7.0%) 11 (8.5%)	0.786
Hospitals of prescription			
Primary Referral centers	2953 (91.0%) 1888 (92.6%)	291 (9.0%) 152 (7.5%)	0.052
Hospitals of monitoring			
Primary Secondary/tertiary	1028 (86.0%) 3813 (93.3%)	168 (14.1%) 275 (6.7%)	<0.001
Indications for PPS use			
Interstitial cystitis Other cystitis Others	4501 (91.6%) 126 (96.2%) 214 (89.2%)	412 (8.4%) 5 (3.8%) 26 (10.8%)	0.066
Mean duration (month)	4.9 ± 8.2	4.9 ± 7.7	0.997
Mean daily dose (mg)	241.8 ± 66.3	247.4 ± 63.5	0.092

PPS, pentosan polysulfate; OCT, optical coherence tomography; FAF, fundus autofluorescence.

**Table 5 jcm-13-05090-t005:** Multivariate logistic regression analysis to identify factors associated with baseline examination (within 1 year) and subsequent and recent monitoring examinations.

Characteristics	Baseline Examination		Subsequent Monitoring		Recent Monitoring	
	OR (95% CI)	*p*-Value	OR (95% CI)	*p*-Value	OR (95% CI)	*p*-Value
Age (years)	1.023 (1.021–1.024)	<0.001	1.023 (1.020–1.026)	<0.001	0.998 (0.995–1.000)	0.090
Sex						
Male Female	1 (reference) 1.279 (1.215–1.346)	<0.001	1 (reference) 0.958 (0.888–1.034)	0.272	1 (reference) 1.038 (0.965–1.117)	0.316
Residence						
Metropolitan area/large cities Small cities/rural	1 (reference) 0.928 (0.885–0.973)	0.002	1 (reference) 1.060 (0.988–1.138)	0.105	1 (reference) 0.891 (0.833–0.954)	0.001
Medical specialties prescribing PPS						
Urology Internal medicine OB-GYN Others	1 (reference) 0.880 (0.757–1.024) 0.806 (0.729–0.890) 0.807 (0.685–0.952)	<0.001	1 (reference) 1.026 (0.820–1.283) 0.998 (0.857–1.163) 1.034 (0.812–1.316)	0.989	1 (reference) 1.000 (0.807–1.241) 1.146 (0.991–1.326) 1.151 (0.909–1.458)	0.208
Hospitals of prescription						
Primary Referral centers	1 (reference) 0.995 (0.944–1.048)	0.849	1 (reference) 1.051 (0.973–1.135)	0.204	1 (reference) 1.069 (0.993–1.151)	0.075
Indications for PPS use						
Interstitial cystitis Other cystitis Others	1 (reference) 0.983 (0.849–1.137) 0.937 (0.831–1.057)	0.565	1 (reference) 1.033 (0.827–1.290) 0.977 (0.814–1.173)	0.928	1 (reference) 0.949 (0.767–1.176) 1.139 (0.953–1.360)	0.308
Mean duration (month)	1.012 (1.009–1.016)	<0.001	1.004 (1.000–1.009)	0.062	0.985 (0.980–0.989)	<0.001
Mean daily dose (mg)	1.000 (1.000–1.000)	0.800	1.000 (0.999–1.000)	0.347	1.000 (0.999–1.000)	0.849

OR, odds ratio; CI, confidence interval; PPS, pentosan polysulfate.

## Data Availability

Data are unavailable due to privacy and ethical restrictions.

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
