# Peer review of "Real-World Practices of Pentosan Polysulfate Maculopathy Screening in Various Clinical Settings and Practice-Associated Factors"

_jcm, 2024, doi:10.3390/jcm13175090_

Round 1

Reviewer 1 Report

Comments and Suggestions for Authors

Dear Authors

You have written an interesting paper. However, the manuscript needs to be improved for greater clarity. The authors are requested to solve the following points:

Abstract

In accordance with the instructions for authors the abstract should consist of about 250 words. The abstract needs to be shortened.

 Introduction

 A period (“.”) at the end of a sentence should be placed after a square bracket with reference number, for example: …[1]. (applies to the entire text of the manuscript) – Instructions for Authors.

The aim of the study should be clearly extract from the introduction, it should be a separate section of the manuscript.

Please complete the data on the pathophysiology of PPS maculopathy.

 Methods

 Please explain what statistical tests were used to test the normality of data and in which cases the results were given as medians (Q1–Q3) or means ±(SD).

Please calculate effect size and confidence interval for effect size. This is essential to better understand the results. Supplementing statistical calculations will affect the clinical  relevance of the conclusions.  

 Results

Please present p-value and effect size results in tables, under the figures or describe in the text. Each p-value must be accompanied by effect size.

 Discussion

Please consider limitations as a separate section of the manuscript.

 Conclusions

The conclusions have been formulated correctly, in the form of implications and suggestions for use in clinical practice and regarding further research opportunities. However, please take into account the possible influence of the supplemented effect size ​​on the obtained results and conclusions.

 References

References No. 2 and 14 are out of date - please replace with newer ones.

Author Response

Author’s Responses to Reviewer’s Comments

Reviewer #1

Comment 1:

You have written an interesting paper. However, the manuscript needs to be improved for greater clarity. The authors are requested to solve the following points:

Abstract

In accordance with the instructions for authors, the abstract should consist of about 250 words. The abstract needs to be shortened.

Response 1:

Thank you very much for your thoughtful review and helpful suggestions. We have shortened the abstract to 250 words by removing unnecessary sentences, in compliance with the journal’s guidelines.

Comment 2:

Introduction

"A period (“.”) at the end of a sentence should be placed after a square bracket with reference number, for example: …[1]. (applies to the entire text of the manuscript) – Instructions for Authors."

Response 2:

We appreciate your comment on the journal instructions. We have revised the manuscript to ensure that periods are placed after the square brackets with reference numbers throughout the text, as per the journal’s guidelines.

Comment 3:

The aim of the study should be clearly extracted from the introduction; it should be a separate section of the manuscript.

Response:

Thank you for the suggestion. Having gone through the journal guideline and already published papers in Journal of Clinical Medicine, ‘Aim of the study’ or ‘Aim’ is not presented as a separate section just as ‘introduction’ and ‘methods’. However, we agree with your point that this should be clearly presented in the introduction section and thus, we have revised the Introduction to include a separate paragraph starting with ‘This study aimed to’, improving the clarity of the aim of the study and also that of manuscript. (Lines 61-68)

The last paragraph of the Introduction section now reads as follows:

This study aimed to explore screening practices among patients administered with PPS in diverse clinical settings, along with the demographic and clinical factors associated with retinopathy screening practices, to provide insights into real-world practice patterns for PPS maculopathy. Accordingly, we investigated the demographic and clinical characteristics associated with PPS maculopathy screening practices in a nationwide population-based cohort in Korea. By reflecting real-world practice patterns across all primary and referral centers at the national level, this study addresses the factors associated with practice patterns, which should be considered to enhance these practices.

Comment 4:

Please complete the data on the pathophysiology of PPS maculopathy.

Response 4:

We appreciate your feedback. We have added sentences on the pathophysiology of PPS maculopathy, which has not been extensively studied, to provide a more comprehensive background for the study, by adding the sentences “PPS maculopathy is characterized by retinal pigment epithelium (RPE) dysfunction and progressive degeneration, often presenting as pigmentary changes and macular atrophy, even after discontinuation of the drug[4-6]. The proposed pathophysiology involves RPE toxicity, possibly associated with PPS disrupting the interphotoreceptor matrix[7].” (Lines )

Comment 5:

Methods

Please explain what statistical tests were used to test the normality of data and in which cases the results were given as medians (Q1–Q3) or means ± (SD).

Response 5:

Thank you for pointing this out. We have added an explanation of the statistical tests used to assess data normality and clarified the criteria for presenting results as means ± SD and median. We apologize for the oversight regarding Q1-Q3, which were not used in our manuscript. We have now included the sentence “Based on the normality of the data, as determined by the Shapiro-Wilk test, either a Student's t-test or a Mann-Whitney U test was used to compare continuous variables between independent groups.” (Lines 127-129). In addition, we corrected the relevant sentence to “continuous variables are reported as means ± standard deviation (SD). We also report the median for the time period due to its skewed (non-normal) distribution.” (Lines 125-127).

Comment 6:

Please calculate effect size and confidence interval for effect size. This is essential to better understand the results. Supplementing statistical calculations will affect the clinical relevance of the conclusions.

Response 6:

Thank you for the suggestion. Odds ratios, with 95% confidence intervals, are commonly used in epidemiological studies and logistic regression to quantify the effect size. Therefore, we have added the sentence “Multivariate logistic regression was performed to identify demographic and clinical factors associated with baseline performance and monitoring examinations. Odds ratios (ORs) were used to quantify the strength of association between variables, with 95% confidence intervals (CIs) provided as a measure of effect size.” in the text. (Lines 130-134) In the Results section, we included 95% CIs for all the ORs.

Comment 7:

Results

Please present p-value and effect size results in tables, under the figures, or describe them in the text. Each p-value must be accompanied by effect size."

Response:

Thank you for the suggestion. We have now included the p-values and corresponding effect sizes (95% CI) with ORs in the tables, figures, and text, where applicable, to ensure a thorough presentation of the results.

Comment 8:

Discussion

Please consider limitations as a separate section of the manuscript.

Response 8:

Thank you for the suggestion. We have created a separate section to address the limitations of our study as per your suggestion.

Comment 9:

Conclusions

The conclusions have been formulated correctly, in the form of implications and suggestions for use in clinical practice and regarding further research opportunities. However, please take into account the possible influence of the supplemented effect size on the obtained results and conclusions.

Response 9:

We appreciate your positive feedback on our conclusions. We have revised the Conclusions section to reflect the potential influence of the effect sizes on our results and their implications for clinical practice and future research. Now the conclusion section reads as:

In summary, the results of this study revealed significant variations in PPS maculopathy screening practices across different settings and medical specialties. These findings emphasize the importance of targeting older adults, particularly women, for baseline and ongoing monitoring. Urologists, who are more frequently involved in PPS prescription, demonstrated a higher adherence to screening practices compared with other medical specialties, indicating a need for enhanced education and guideline dissemination among those other specialties, particularly OB/GYN. The observed patterns in screening practices highlight the need for tailored and proactive monitoring strategies. While the effect sizes are relatively small in this study, their implications for clinical practice should not be overlooked, particularly in guiding targeted interventions. Future research should validate findings across diverse populations and refine screening protocols to ensure early detection and management of PPS-related maculopathy. By improving awareness and implementing targeted screening practices, healthcare providers can better manage the risks associated with PPS therapy and enhance patient outcomes. (Lines 368-381)

Comment 10:

References

"References No. 2 and 14 are out of date - please replace them with newer ones."

Response 10:

Thank you for noting this. We have updated References 2 and 14 with more recent sources (published in 2015 and 2021, respectively) to ensure that our manuscript reflects the most current research in the field.

Reviewer 2 Report

Comments and Suggestions for Authors

Dear Authors,

I wish to submit my review of the paper: " Real-World Practices of Pentosan Polysulfate Maculopathy Screening in Various Clinical Settings and Practice-associated  Factors"

The paper explores a novel topic and is interesting and commendable in its analysis. The Authors should be commended for their work. However, some points need attention.

The introduction should be amended and expanded to highlight the features of Pentosan Polysulfate Maculopathy and report the current practice algorithm. Regarding the statistical analysis, the paragraph is concise. Indeed, the Authors should report how they assessed the data normality and the tests used to analyze quantitative variables. Have the Authors performed multivariate or multivariable regression?

The total number of included patients reported in Figure 1 should be reported in the main text (Result section). The discussion section should be amended and expanded to discuss the provided results further critically. In addition, the Authors should state how the presented data can significantly impact current and future clinical practice/healthcare systems and how this research can be a valuable resource for clinicians in their daily practice.

Comments on the Quality of English Language

 Minor editing of English language required

Author Response

Reviewer #2

Comment 1:

I wish to submit my review of the paper: "Real-World Practices of Pentosan Polysulfate Maculopathy Screening in Various Clinical Settings and Practice-associated Factors"

The paper explores a novel topic and is interesting and commendable in its analysis. The Authors should be commended for their work. However, some points need attention.

The introduction should be amended and expanded to highlight the features of Pentosan Polysulfate Maculopathy and report the current practice algorithm.

Response 1:

Thank you for your review and valuable comments. We have revised the Introduction to include a detailed description of pentosane polysulfate maculopathy, highlighting its key features and current practice recommendations. This addition may provide a more comprehensive background and enhance readers’ understanding of our study.

In the text, we have added the following sentences:

PPS maculopathy is characterized by retinal pigment epithelium (RPE) dysfunction and progressive degeneration, often presenting as pigmentary changes and macular atrophy even after drug[4-6]. The proposed pathophysiology involves RPE toxicity, possibly associated with disruption of the interphotoreceptor matrix[7]. This condition is critical because it can lead to permanent vision loss[4,7]. The prevalence of PPS-associated maculopathy is increasing, and studies have indicated a potential risk for chronic PPS users[7,8]. Advanced imaging modalities such as optical coherence tomography (OCT), near-infrared reflectance (NIR), and fundus autofluorescence (FAF) facilitate the early detection of this condition, enhancing the ability to prevent significant visual impairment through timely drug cessation[7]. (Lines 37-46)

Some authors recommend an initial baseline eye examination for patients starting PPS therapy, followed by annual eye examinations using multiple imaging techniques, such as OCT, FAF, and NIR, for those with chronic PPS exposure (e.g., cumulative dosages exceeding 500 g)[7]. (Lines 49-52)

Comment 2:

Regarding the statistical analysis, the paragraph is concise. Indeed, the Authors should report how they assessed the data normality and the tests used to analyze quantitative variables. Have the Authors performed multivariate or multivariable regression?

Response 2:

We appreciate your comments in the Statistical Analysis section. Per your suggestion, we have included details on how data normality was assessed, the statistical tests used to analyze the quantitative variables, and how multivariate regression analyses were conducted. In the text, we have added the sentences “Based on the normality of the data, as determined by the Shapiro-Wilk test, either a Student's t-test or a Mann-Whitney U test was used to compare continuous variables between independent groups.“ (Lines 127-129) and “Multivariate logistic regression was performed to identify demographic and clinical factors associated with baseline performance and monitoring examinations. Odds ratios (ORs) were used to quantify the strength of association between variables, with 95% confidence intervals (CIs) provided as a measure of effect size.” (Lines 130-134)

Comment 3:

The total number of included patients reported in Figure 1 should be reported in the main text (Results section).

Response 3:

Thank you for the suggestion. We have included the total number of patients depicted in Figure 1 in the Results section to ensure that the study population is clearly presented. In the Results section, the sentence “Table 1 provides an overview of the demographic and clinical characteristics of 58,752 patients using PPS included in this study.” has been added to the first sentence. (Lines 139-140)

Comment 4:

The discussion section should be amended and expanded to discuss the provided results further critically. In addition, the Authors should state how the presented data can significantly impact current and future clinical practice/healthcare systems and how this research can be a valuable resource for clinicians in their daily practice.

Response 4:

We agree with your observation that the Discussion section needs further elaboration. We have expanded this section to provide a more critical analysis of the results including medical specialties. Additionally, we have discussed how our findings could significantly impact current and future clinical practices and healthcare systems. Finally, we have highlighted the potential value of our research as a resource for clinicians.

In the text, we have added the following paragraph “The disparity in screening practices among different specialties highlight a critical gap that could significantly impact both current and future clinical practice of maculopathy screening. Addressing this gap through targeted interventions could help standardize care, ensuring that all PPS users receive appropriate and timely screening, regardless of the prescribing medical specialty. For example, targeted educational initiatives and enhanced communication between prescribing and screening physicians are crucial to ensuring that OB/GYNs are equally informed and proactive in referring patients for PPS-related maculopathy, ultimately improving patient care and outcomes. Furthermore, this research serves as a valuable resource for clinicians by underscoring the importance of cross-specialty education and collaboration to improve patient outcomes and ensure ocular safety in PPS users. Incorporating these findings into clinical guidelines and healthcare policies across multiple specialties, similar to the approach taken with hydroxychloroquine retinopathy[20], could lead to more consistent and effective maculopathy screening practices.” (Lines 333-346) and also the sentence “Nevertheless, these findings highlight the importance of tailored and ongoing monitoring strategies, particularly for long-term PPS users, to ensure timely, regular screening and address potential gaps in follow-up care for maculopathy despite a higher risk of maculopathy.” (Lines 320-323)
